# Exploration of Artificial Intelligence Use with ARIES in Multiple Myeloma Research

**DOI:** 10.3390/jcm8070999

**Published:** 2019-07-09

**Authors:** Sophia Loda, Jonathan Krebs, Sophia Danhof, Martin Schreder, Antonio G. Solimando, Susanne Strifler, Leo Rasche, Martin Kortüm, Alexander Kerscher, Stefan Knop, Frank Puppe, Hermann Einsele, Max Bittrich

**Affiliations:** 1Department of Internal Medicine II, University Hospital Würzburg, 97080 Würzburg, Germany; 2Chair for Artificial Intelligence and Applied Informatics, University of Würzburg, 97070 Würzburg, Germany; 3I. Medical Department, Wilheminen Hospital, 1160 Vienna, Austria; 4Department of Biomedical Sciences and Human Oncology, Section of Internal Medicine ‘G. Baccelli’, University of Bari Medical School Bari, 70124 Bari, Italy; 5Comprehensive Cancer Center, 97080 Würzburg, Germany

**Keywords:** natural language processing, ontology, artificial intelligence, multiple myeloma, real world evidence

## Abstract

Background: Natural language processing (NLP) is a powerful tool supporting the generation of Real-World Evidence (RWE). There is no NLP system that enables the extensive querying of parameters specific to multiple myeloma (MM) out of unstructured medical reports. We therefore created a MM-specific ontology to accelerate the information extraction (IE) out of unstructured text. Methods: Our MM ontology consists of extensive MM-specific and hierarchically structured attributes and values. We implemented “A Rule-based Information Extraction System” (ARIES) that uses this ontology. We evaluated ARIES on 200 randomly selected medical reports of patients diagnosed with MM. Results: Our system achieved a high F1-Score of 0.92 on the evaluation dataset with a precision of 0.87 and recall of 0.98. Conclusions: Our rule-based IE system enables the comprehensive querying of medical reports. The IE accelerates the extraction of data and enables clinicians to faster generate RWE on hematological issues. RWE helps clinicians to make decisions in an evidence-based manner. Our tool easily accelerates the integration of research evidence into everyday clinical practice.

## 1. Introduction

Multiple Myeloma (MM) is the third most common hematological malignancy in Germany [1]. There is still no curative MM treatment, but new drug combinations and therapy protocols are improving overall survival. However, to investigate such new drugs it is necessary to have as much data as possible. One source is routinely collected patient information. Unfortunately, it is a laborious and time-consuming task to extract data from this source for Real-World Evidence (RWE) analysis. RWE is defined as “the technology-facilitated collation of all routinely collected information on patients from clinical systems to a comprehensive, homogeneously analyzable dataset (big data) that reflects the treatment reality in the best possible and comparable manner” [2]. Extraction is even more difficult in retrospective data collection, which is often based on unstructured texts like discharge letters or medical records [3].

Doctors at university hospitals are researchers and clinicians at the same time. Under some circumstances, they are able to offer new drugs or new combination regimens in therapy. Some of these may be awaiting approval by European or local regulatory authorities. But, even when patients do not meet the criteria for inclusion in a clinical trial, clinician researchers can nevertheless offer the therapy if, on medical request, the health insurance agrees to cover the costs. The patients then receive up-to-date treatment.

As a consequence, scientific information is not always collected within the scope of clinical trials. RWE is therefore of high importance. Furthermore, the results of clinical trials mirror the real world to only a limited degree because they are highly regulated e.g., by very strictly selecting patients. Therefore, it is essential to generate data through the RWE approach. RWE is increasingly necessary for academic questions, (post-) approval processes and complements randomized controlled trials [2]. It is of great interest to researchers for accelerating the process of retrieving data, especially for retrieving data from unstructured texts [3].

This process of retrieving data can be supported by Artificial Intelligence (AI). We approached this problem by implementing a rule-based information extraction (IE) system. This tool is specialized for retrieving information from heterogeneously structured discharge letters and written medical reports in the German language. The IE algorithm is able to collect not only patient characteristics but even information that is specific to plasma cell disorders. For example, the algorithm is able to retrieve the different stage classifications, laboratory parameters, therapy protocols, responses to therapy, adverse reactions and comorbidities. 

AI in the form of natural language processing (NLP) has several fields of application in clinical medicine. For example, NLP accelerates the translation of cancer treatments from the laboratory to the clinic [3]. In this regard, a crucial question is how NLP is able to generate information out of unstructured written texts [4]. Examples of IE can be found in radiology reports [5] or transthoracic echocardiography reports [6]. Moreover NLP finds application in oncology for case identification and for disease stages and outcome determination [3]. NLP has been used to query clinical reports on MM-specific parameters [7]. However, our ontology provides much more extensive querying than these trials [7]. In addition, the extraction of information in the German language is still very limited [8].

The aim of our IE tool is to quickly and easily enable RWE extraction for different treatments in hematology and for further medical questions. 

## 2. Methods

### 2.1. A Rule-Based Information Extraction System

We implemented ARIES, which relies on predefined ontologies to extract data from unstructured text. ARIES is therefore an ontology-based IE system [9]. 

Our ontologies consist of two types of classes: attributes and values. Attributes are the main parameters extracted from the text. Attributes include, for example, diagnoses and medications in discharge letters or anatomical findings in radiology reports. Values, however, are more descriptive features of attributes and therefore only exist in conjunction with an attribute. Values include, for example locations, the severity of an attribute or the date and time at which the attribute is measured. Since most values obviously are relevant for more than one attribute, they are organized in groups, which we refer to as templates. Templates can be treated as multiple or single choice groups, for example, a tumor can have multiple locations, but only one date of primary diagnosis. An attribute can have several templates assigned. 

Every attribute and value in the ontology contains extraction rules in the form of regular expressions (regex) for finding corresponding entities in a given text. Regex are sequences of characters describing a search pattern. Regex are used to find matching parts in the document. In addition to naive search functions, regex implement improved functions like wildcards, repetitions of substrings and additional optional parts. In the first step, the text is split into segments with a configurable splitting function. That allows the definition of a regex as a splitting point. For example, if the regex matches on periods, an abbreviation detection mechanism is applied to prevent the system from splitting sentences too early. In the second step, all defined attributes and values are marked for every segment. Attributes can exist without further values, so they always get extracted. Values are discarded if they are without at least one corresponding attribute according to the assignment in the ontology. Values that have more than one possible match are linked to the nearest logical neighbor attribute. When the system is trying to find single choice templates, only the longest match out of all possible values is considered. Furthermore, every attribute can have a numerical value, for example, laboratory values. A numerical attribute is defined by giving it a corresponding unit, for example ‘cm’ centimeters for height. If there are different possible units, the ontology editor can define conversion factors between (e.g., “m” in “cm”). The values of all processed instances are subsequently normalized to a uniform target unit. If there are multiple numerical expressions in the same segment as the attribute, the nearest logical neighbor candidate is again chosen. Figure 1 shows the architecture of the ARIES algorithm. 

For editing the ontology, we used webATHEN (A public beta version of webATHEN is available under http://webathen-beta.informatik.uni-wuerzburg.de/), a web clone of ATHEN [10]. This webATHEN editor includes the ARIES algorithm. It is therefore possible to edit the ontology and execute it on sample documents on-the-fly during its development. We used this tool because it follows a simpler approach than, for example, Protégé [11]. So, domain experts or researchers, even if they are not computer scientists, can operate webATHEN without extensive training periods.

ARIES is implemented in Kotlin and Java and is compatible with the UIMA Framework. Therefore, it can easily be used in other UIMA based IE environments like cTakes [12], given an existing ontology. 

### 2.2. Multiple Myeloma Ontology

In the MM ontology the attributes were hierarchically structured into parent and child attributes. The parent attributes are “CRAB”, “Cytogenetic”, “Laboratory”, “Myeloma specific”, “Adverse Events, Complications, Comorbidities”, “Response Criteria”, “Risk Factors”, “Staging”, “Therapy” and “Substances”.

Adding child attributes structures the ontology. For example, the attribute “Cytogenetic” has the child attribute “High Risk Cytogenetic” which includes the following for attributes relevant to MM: “del(17/17p)”, “t(4;14)”, “t(14;16)”, “t(14;20)” and “gain(1q)”. Appendix A shows all extractable attributes of our ontology. Multiple synonym expressions were added for every attribute. This ensures the correct extraction in spite of the unstructured and free text written by different authors.

Regex were used to generalize the synonyms of an attribute and to achieve a high hit rate. For the attribute “del(17/17p)” some synonyms are “del17p”, “17p-Deletion”, “TP.53.Deletion”, “del.*17p13.*”. In this way, as many as possible synonyms of an attribute are included in the IE terminology, representing the alternative spellings used in written language.

Regex were furthermore used to differentiate between attributes. For example, there are several laboratory parameters for lambda or kappa light chains. This laboratory parameters are similar in spelling but need to be distinguished. It is important to differentiate the parameter “kappa Leichtk”, which measures the kappa light chains in the serum, from the parameter “kappa Leichtk. (U)”, which measures the kappa light chains in the urine of a patient. Therefore the regex negative lookahead “kappa. Leichtk.(?!(.*170 - 370|.*\(PU\)|.*\(U\)))” was included as a synonym of the parameter “kappa-Leichtk.” Negative lookahead means the attribute cannot be followed by the characters in the brackets. Thus the terminology does not read out the attribute “kappa Leichtk.” if this term is followed by “(U)”.

The ontology includes templates for the concepts “Bisphosphonate Substance”, “Date”, “Adverse Events”, “Grade of AE”, “ICD-10”, “Light Chains”, “Light Chain Myeloma”, “Heavy Chains”, “Protocol”, “Response”, “Therapy”, “Salmon and Durie”, “ISS” and “R-ISS”. These templates contain several template values such as “I”, “II” and “III” for the (Revised-) International Staging System “ISS” or “R-ISS”. In this example “ISS” is a single choice template. Therefore, the ontology will only read out the template “III” for the attribute “III” despite of the term “III” containing “I” and “II”.

### 2.3. Dataset

We tested our system on the medical reports of 261 patients who met the inclusion criteria. We included only participants who were 18 years or older, suffered from MM, received a quadruple combination treatment and for whom data was fully available in the hospital information system. These quadruple combinations were “VRCd” (Bortezomib, Lenalidomide, Cyclophosphamide, Dexamethasone), Pom-PAd (Pomalidomide, Bortezomib, Doxorubicin, Dexamethasone) and PAd-Rev (Bortezomib, Doxorubicin, Dexamethasone, Lenalidomide). The exclusion criteria were patients younger than 18 years, patients with incomplete data sets, patients who did not give consent or withdrew consent, or patients who did not receive any of the therapies mentioned above. For these patients, we retrieved 5456 anonymized discharge letters in the German language from the Clinical Data Warehouse (CDW) at the University Hospital of Würzburg in Germany, a reference center for MM. Table 1 shows a description of our patient cohort. This cohort is representative of myeloma cohorts described elsewhere for Germany, with 64% being male and a mean age at diagnosis of 58 years. All patients had relapsed or refractory MM (RRMM).

For developing an initial version of the ontology, 100 randomly selected medical reports were used as the training set. To improve the algorithm and ontology, 100 more documents were randomly sampled and annotated with correct gold annotations as the validation set. Another 100 randomly selected letters were also annotated with correct gold annotations but were retained as an unseen evaluation set.

### 2.4. Ethical Approval 

This study obtained ethical approval in 2013 from the local ethics committee of the University of Würzburg (reference number 76/13). 

## 3. Results

After optimizing the ontology on the training and validation set, an evaluation run was performed on the evaluation set. The common measures in NLP—precision, recall and F1 score—were calculated (Table 2).

Overall, the system achieved an F1-Score of 0.92 on the evaluation dataset which is slightly worse than the result from the training dataset (0.93). This is expected because there are always unseen cases in retained datasets. The system achieved the same precision on both evaluation and training data (0.87). The recall of 0.98 is very high on the evaluation dataset and even higher on the training data (0.99).

Table 3 shows the results of a few chosen attributes that are of special medical relevance. Some parameters were very reliably extracted, whereas others were not. For example, the extraction of the “ISS stage III” flawlessly worked (F1 1.0), while the extraction of “Light chain type kappa” suffers from a very low precision of 0.33. All the attributes chosen had very high recall and eight out of the ten chosen attributes had a F1-Score of 0.9 or higher. 

A more detailed error analysis for the false negatives shows Table 4. In total, the IE tool produced 298 false-negative results. This number was made up of 181 values, 112 attributes and five incorrectly extracted relations. The majority of mistakes were missing values due to a missing attribute, which occurred 82 times. 76 out of these 82 Errors were missing synonyms of attributes but only three were missing synonyms of values. 20 values and 15 attributes were not retrieved because of imprecise wording in the written medical reports. In addition, 29 values and one attribute were not extracted because of ambiguous meanings. False-negative mistakes were less common due to spelling errors in the written medical reports. These spelling errors occurred 17 times with attributes and six times with values. A minor role played errors regarding the single check option, the segmentation or wrong or missing units. 

## 4. Discussion

Written medical reports such as discharge letters or tumor conference protocols are a way of communication in everyday clinical practice [3]. These reports are used by treating doctors of different departments or are sent to the general practitioners (GPs) of patients in order to update the GPs on the medical history and condition of shared patients. In addition, these reports record information for subsequent consultation. Thus, written medical reports present a large amount of clinical real world data which is available for retrospective research [2]. It is however a laborious and time-consuming task to tap into this valuable RWE information. One reason is that only a small part of any medical report is structured [3]. Reports often change in structure over time or because they have different authors. Unstructured and imprecise written medical reports are also often produced because physicians have little time and high workloads. While some report passages are written in a staccato-like noun phrase style, they also contain lots of written unstructured text. This relevant data in the report is less accessible and more difficult to filter out. As a result, a lot of information is lost in incoherent reports.

Acquiring RWE is of utmost importance today [2,3,13]. It takes on a strong complementary position to randomized controlled trials (RCT) with their limitations. RCTs are still the gold standard of research but they are limited in their generalizability and applicability to everyday clinical practice [13]. Only 3–5% of cancer patients receive treatment as part of an RCT [13,14]. For example, RCT are conducted on patients that are much younger and have fewer comorbidities than the average oncological patient [14,15]. Moreover, RCTs are high in cost and expenses, whereas RWE can tap into already existing and available clinical data.

While therapeutic options and the demand for individual medicine increase, there is a need for a tool that improves oncological evidence-based research and quality [3]. NLP can accelerate the process of translating cancer treatment from bench to bedside or from academic centers to being a reality of everyday treatment. NLP also supports generating RWE. RWE helps clinicians to make decisions in an evidenced-based manner. Thus, NLP helps in integrating research into everyday clinical practice [3]. First, NLP can harvest valuable information which would otherwise be lost in free text medical reports. Second, NLP saves resources and accelerates the querying of medical reports and the retrieval of information [3]. 

Our objective was therefore to support the process of retrieving this data out of retrospective clinical written reports in the German language. The emphasis was hereby on extracting clinical and research relevant information on patients diagnosed with MM. Our tool queries a vast amount of MM related attributes, as shown in Appendix A. It is constructed to extract the baseline characteristics such as specific laboratory parameters or staging classifications. Furthermore, it is trained to capture the different treatments and therapy protocols, the responses to them, and the adverse events or relapses that occur. However, the extraction is not limited to this. It is able to retrieve comprehensive data on each individual history. Examples are patient risk factors or comorbidities. Our ontology can easily be extended by attributes, expanding its range of query even further. Our current system is only able to query reports written in German. However, some attributes are the same in English, whereas others can be easily translated and added. 

The ontology created is particularly appropriate for supporting retrospective studies of MM protocols in regard to overall response (ORR), progression free survival (PFS) and overall survival (OS). We are in the process of using our terminology to perform a study on the effectiveness, ORR, PFS and OS of myeloma patients who received a quadruple therapy.

NLP has a wide range of application in research and everyday clinical practice. It is of use for example in radiology reports [5] or transthoracic echocardiography reports [6]. In oncology NLP is used for identifying disease stages [16], and evaluating outcomes (such as adverse drug reactions [17] or tumor recurrences [18]). Other NLP projects targeting MM take into account performance to identify genes associated with MM [19], “a standardized hierarchic ontology of cancer treatments” [20] or an “ontology-driven semiological rules base and a consultation form to aid in the diagnosis of plasma cells diseases” [21].

Nevertheless, we only found one case in which NLP was utilized to retrieve patient data from unstructured free text on MM. Löpprich, Krauss et al. [7] created a “multiclass classification of free-text diagnostic reports” and “a framework to enable automatic multi-class classification of relevant data elements from free-text diagnostic reports”. Their aim was to automatically document “diagnosis and state of disease of myeloma patients” from clinical reports. They trained two different classifiers—a support vector machine (SVM) and a maximum entropy classifier (MEC)—on a self-created dataset and obtained F1 scores: 0.92 for SVM, 0.89 for MEC, similar or slightly inferior to ours. However, their range of parameters was limited and their data source was only a small part of the whole document (the main diagnosis paragraph). Our tool thus exceeds theirs in the range of myeloma-specific and additional data queried.

One possible limitation of our system is that it has a lower precision than recall. Our precision in the evaluation was 0.87, while the recall amounted to 0.98. We developed our system in a way, that attributes were correctly identified and were only extracted if identified in the right context. There were a lot of false positives from attributes that were correctly extracted but not found in the right context. For example, every discharge letter has an introduction instructing the GP to contact the clinic “if the patient has fever over 37.5 degree Celsius”. The algorithm correctly finds the attribute “fever” and relates it to the value “37.5” and unit “Degrees Celsius”, but this is not the right context because it is just a recommendation and not a real status. If a researcher is looking for patients suffering from “fever” in a CDW or cancer registry, he would find this patient mistakenly. Another example is the extraction of “light chain type kappa”, which suffers from a low precision (Table 3). This low precision arises from our very strict evaluation. We considered the data to be correctly annotated only if it was extracted in the right context. Since “kappa” is frequently used and is also a component of multiple parameters of our MM ontology, we received a high number of false positives. These redundantly extracted attributes and values are easily understood as “not relevant” by a person manually working in a medical field. But automatic data mining struggles with such text. Nevertheless, several of the attributes chosen (Table 3) reached a high precision and F1-Score. Especially successfully extracted were “ISS Stage III”, “Bone marrow infiltration”, “Paraprotein” and “Pathological fracture” and this shows the strength and reliability of our tool.

Our error analysis (Table 4) shows that the majority of mistakes were missing values due to missing attributes. Most of these values were not detected because the corresponding attribute was not in the same segment. For example, if the value “VCD”, a therapy protocol abbreviation, stands in one segment without its attribute “Cycle”, it won’t be retrieved. We can solve this problem by adding these values as attributes. The second most frequent error was missing synonyms. These were often associated with a corresponding value and so were not detected either. Missing synonyms of values however were not so much of a problem. Other common errors were values and attributes that were not retrieved due to imprecise wording in the written medical reports. One example is an author who insufficiently describes the Salmon and Durie Staging System. In these cases, reports mentioned the stage without details. In addition, values and one attribute with an ambiguous meaning were not extracted. This problem particularly occurred with abbreviations. One example is the response criterion “stable disease”, which is abbreviated as “SD”. However, “SD” is ambiguous because it is also an abbreviation for the German word for thyroid gland: “Schilddrüse”. 

Another possible limitation is the patient cohort on which we trained the ontology. We didn’t have a detailed look at these patients. We randomly picked medical reports of patients, meeting our inclusion and exclusion criteria. There can be a selection bias regarding this inclusion and exclusion criteria. We will address the details of several patient cohorts with ARIES in future research. 

Our ontology-based tool can be used as a data extraction tool for multiple purposes. This tool is integrated into the CDW of the University Hospital of Würzburg [22]. Extracted data can be combined with other facts from structured data like ICD-10 or OPS codes. The CDW can be used for further integrity checks. One example is the post-processing of “ISS” and “R-ISS” stage, which can be checked for reliance. It is not logically possible to have the stage “ISS” with value “I” together with “R-ISS” value “III”. Such an occurrence indicates an error in the report or the extraction. Several criteria have to be fulfilled for the “R-ISS” stage “III”, including “ISS” stage “III” and “above upper limit lactate dehydrogenase in laboratory” or “one high risk marker present in cytogenetic”.

Moreover, our ontology is implemented in the cancer registry of lower Franconia, a part of the cancer registry in Bavaria. CDWs and cancer registries contain big data and bring together a great deal of important information in one place. Big data analysis and results should be integrated into research and clinical practice. Access to big data is of great relevance for cancer research and will help clinicians evaluate outcomes, make evidence-based decisions and promote personalized medicine, especially in academic centers where many uncommon therapies are applied [3,23,24,25]. 

Because the data extracted for the cancer registry will be corrected later on, we are planning to use our system to build a larger gold dataset for the above defined attributes. With a larger dataset, we will reproduce, or even outperform, the results with a deep learning approach or other machine learning algorithms, such as neural networks.

## 5. Conclusions

We demonstrate the creation of the first comprehensive rule-based IE system able to query unstructured written medical reports in MM. While our tool has slightly lower precision than recall, the redundantly extracted information is recognized as irrelevant by medical professionals. Our system is better than existing ones in extracting MM-specific information. It is designed to query German medical reports, however it is easily possible to add translations into English or other languages. The main focus was to address plasma cell disorder issues, thus helping to generate RWE. RWE helps clinicians to decide in an evidence-based manner and integrates research into everyday clinical practice. We developed an IE system which enables and accelerates the extraction of valuable patient information. This will allow us to generate big data for clinical trials, cancer registries, and for future machine learning approaches.

## Figures and Tables

**Figure 1 jcm-08-00999-f001:**
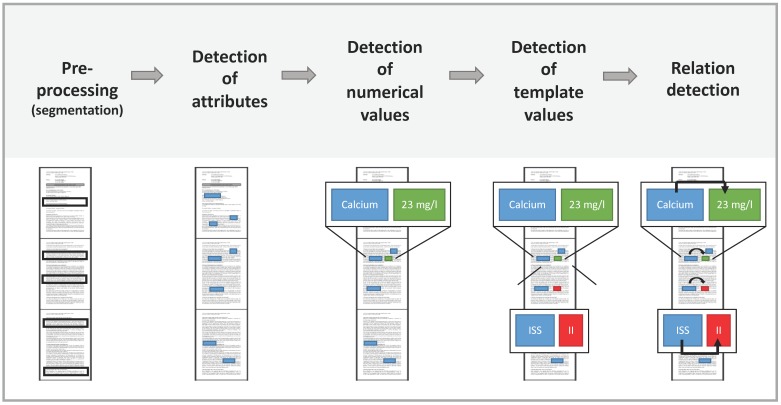
Architecture of the ARIES algorithm. After pre-processing ARIES detects attributes, numerical values and template values. Next, ARIES relates the detected information.

**Table 1 jcm-08-00999-t001:** Patient characteristics of our cohort.

Variable	All Patients (*n* = 261)
Sex	
Female	95
Male	166
Age at diagnosis, years (range)	58 (33–82)
Therapy	
VRCd ^1^	74
Pom-Pad ^2^	123
PAd-Rev ^3^	64
RRMM ^4^	261

^1^ VRCd: Bortezomib, Lenalidomide, Cyclophosphamide, Dexamethasone; ^2^ Pom-PAd: Pomalidomide, Bortezomib, Doxorubicin, Dexamethasone; ^3^ PAd-Rev: Bortezomib, Doxorubicin, Dexamethasone, Lenalidomide; ^4^ RRMM: relapsed or refractory Multiple Myeloma (MM).

**Table 2 jcm-08-00999-t002:** Results of the information extraction (IE) system on training and evaluation data. We counted every attribute-value combination as well as single attribute without a value as an entity.

	TP	FP	FN	Precision	Recall	F1
**Training**	16888	2517	213	0.87	0.99	0.93
**Evaluation**	16077	2442	298	0.87	0.98	0.92

TP: True Positive; FP: False Positive; FN: False Negative.

**Table 3 jcm-08-00999-t003:** Detailed results for selected MM-specific parameters.

	TP ^1^	FP ^2^	FN ^3^	Precision	Recall	F1
Del(17/17p)	25	0	2	1.0	0.93	0.96
ISS stage III	10	0	0	1.0	1.0	1.0
Bone marrow infiltration	206	2	8	0.99	0.96	0.98
M protein level	101	19	0	0.84	1.0	0.91
Paraprotein	106	19	0	0.85	1.0	0.98
Pathological fracture	23	0	1	1.0	0.96	0.98
Serum light chain ratio	95	14	8	0.87	0.92	0.90
Date of primary diagnosis	123	7	0	0.95	1.0	0.97
Heavy chain type IgG	111	42	0	0.73	1.0	0.84
Light chain type kappa	123	246	0	0.33	1.0	0.5

^1^ TP: True Positive; ^2^ FP: False Positive; ^3^ FN: False Negative.

**Table 4 jcm-08-00999-t004:** Results of the detailed error analysis.

Type of Mistake	Missing Value	Missing Attribute	Wrong Relation
Missing synonym	3	76	
Ambiguous meaning	29	1	
Missing value due to missing attribute	82	0	
Imprecise wording	20	15	
Single check option	8	0	
Spelling error	6	17	
Segmentation error	12	0	
Wrong unit	11	0	
Not assignable	10	3	
Total	181	112	5

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
