# Peer review of "Exploration of Artificial Intelligence Use with ARIES in Multiple Myeloma Research"

_jcm, 2019, doi:10.3390/jcm8070999_

Round 1
Reviewer 1 Report
line 59 "might not be admitted yet" difficult to understand
line 62 "Nevertheless, it is essential to generate data out of an RWE approach," suggest therefore
line 77 "NLP has been used in before" in?
avarage reader might need an explanation of what regex is
Author Response
28th of June 2019
Dear reviewer,
Thank you for revising the manuscript and your comments and help.
We tried our best to implement your remarks into our manuscript.
Point 1: English language and style are fine/minor spell check required
Response 1: We had our manuscript revised by Dr. A.J. Davis, English Experience Language Services, Göttingen, Germany. The help of Doctor Davis is stated at the end of the contribution section.
Point 2: line 59 "might not be admitted yet" difficult to understand
Response 2: We tried to explain this better: “Doctors at university hospitals are researchers and clinicians at the same time. Therefore, under some circumstances, they are able to offer new drugs or new combination regimens in therapy. Some of these may be awaiting approval by European or local regulatory authorities. But, even when patients do not meet the criteria for inclusion in a clinical trial, clinician researchers can nevertheless offer the therapy if, on medical request, the health insurance agrees to cover the costs. The patients then receive up‑to‑date treatment.” You can find it in line 166 ff. of the manuscript.
Point 3: line 62 "Nevertheless, it is essential to generate data out of an RWE approach," suggest therefore
Response 3: We exchanged nevertheless for therefore.
Point 4: line 77 "NLP has been used in before" in?
Response 4: The “in” was a spelling mistake, which we corrected.
Point 5: average reader might need an explanation of what regex is
Response 5: We included an explanation of regex: “Regex are sequences of characters describing a search pattern. They are used to find matching parts in the document. In addition to naive search functions, regex implement improved functions like wildcards, repetitions of substrings and additional optional parts..” You can find it in line 342 ff. of the manuscript.
Thank you for revising our manuscript.
Kind regards,
Sophia Loda
Reviewer 2 Report
I presume the dataset included only German patients - were the records written in English or German? I suppose in German - are the results then usable for English? And since all patients originated in one institution, is it possible that the style of the records is similar?
Author Response
28th of June 2019
Dear reviewer,
Thank you for revising the manuscript and your comments and help.
We tried our best to implement your remarks into our manuscript.
Point 1: Extensive editing of English language and style required
Response 1: We had our manuscript revised by Dr. A.J. Davis, English Experience Language Services, Göttingen, Germany. The help of Doctor Davis is stated at the end of the contribution section.
Point 2: Does the introduction provide sufficient background and include all relevant references? Can be improved.
Response 2: We added missing information to the introduction, especially regarding the language and nature of the medical reports as you can see for example in line 181 and 182. We added further information concerning this matter in the methods and discussion setting, please refer response 3 - 5.
Point 3: I presume the dataset included only German patients - were the records written in English or German?
Response 3: The dataset included patients of the University Hospital of Würzburg (l. 182). We included that these records were written in German in line 85 of the introduction, line 502 of the methods section, line 650 of the discussion and line 968 of the conclusion.
Point 4: are the results then usable for English?
Response 4: We added the following to clarify this point: “Our current system is only able to query reports written in German. However, some attributes are the same in English whereas others can be easily translated and added.” (line 658 ff.)
Point 5: And since all patients originated in one institution, is it possible that the style of the records is similar?
Response 5: We included into the introduction that the reports are of very heterogenous structure (line 181). Moreover, we would like to refer to a paragraph in the discussion section, which explains why the style of the records differs, even though all reports are from the same institution (line 629 ff.). We are trying to get anonymized data from other institutions to validate this ontology further.
Thank you for revising our manuscript.
Kind regards,
Sophia Loda
Reviewer 3 Report
In this manuscript, Lodha et.al. have demonstrated the utilization of tools to extract Multiple myeloma-specific ontology from unstructured texts. This group showed the applicability of the ARIES tool to query 200 medical reports. The authors claim that this rule-based query expedites extraction of data thereby providing clinicians with valuable real-world evidence for hematological diseases. The use of an AI-based approach for NLP provides value for extraction of information in the german language.
Overall the manuscript is well written with good attention to grammar and appropriate description of content. There is equal weightage given to the introduction, materials, results and discussion section with good discussions for each section.
The results are well represented and explained in great detail that make it eeasier for the reader to interpret. For the results table, it would be useful to the users if the abbreviations for TP, FP and FN are re-written below the table. The only section that needs a little more elaboration is the conclusions section. Here a restatement of the strengths and weaknesses of the tools presented in this manuscript will strengthen the overall quality of the manuscript.
Author Response
28th of June 2019
Dear reviewer,
Thank you for revising the manuscript and your comments and help.
We tried our best to implement your remarks into our manuscript.
Point 1: English language and style are fine/minor spell check required
Response 1: We had our manuscript revised by Dr. A.J. Davis, English Experience Language Services, Göttingen, Germany. The help of Doctor Davis is stated at the end of the contribution section.
Point 2: For the results table, it would be useful to the users if the abbreviations for TP, FP and FN are re-written below the table.
Response 2: We added a caption to table 2, explaining the abbreviations in line 544.
Point 3: The only section that needs a little more elaboration is the conclusions section. Here a restatement of the strengths and weaknesses of the tools presented in this manuscript will strengthen the overall quality of the manuscript.
Response 3: We included a more elaborate conclusion especially regarding the strengths and weaknesses. You can find these changes in line 964 ff.
Thank you for reviewing our manuscript.
Kind regards,
Sophia Loda
Round 2
Reviewer 2 Report
I have no more comments. The paper has been improved.
Author Response
Dear reviewer,
Thank you for your comments and help.
Kind regards,
Sophia Loda